# An Adaptive Genetic Algorithm of Adjusting Sensor Acquisition Frequency

**DOI:** 10.3390/s20040990

**Published:** 2020-02-12

**Authors:** Feng Chen, Shouzhi Xu, Ying Zhao, Hui Zhang

**Affiliations:** College of Computer and Information Technology, China Three Gorges University, Yichang 443002, China; 201708100021001@ctgu.edu.cn (F.C.); 201708100021012@ctgu.edu.cn (Y.Z.); 201808521121031@ctgu.edu.cn (H.Z.)

**Keywords:** data acquisition, frequency regulation, low power consumption, wireless sensor

## Abstract

Portable meteorological stations are widely applied in environment monitoring systems, but they are always limited in power-supplying due to no cable power, especially in long-term monitoring scenarios. Reducing power consumption by adjusting a suitable frequency of sensor acquisition is very important for wireless sensor nodes. The regularity of historical environment data from a monitoring system is analyzed, and then an optimization model of an adaptive genetic algorithm for environment monitoring data acquisition strategies is proposed to lessen sampling frequency. According to the historical characteristics, the algorithm dynamically changes the recent data acquisition frequency so as to collect data with a smaller acquisition frequency, which will reduce the energy consumption of the sensor. Experiment results in a practical environment show that the algorithm can greatly reduce the acquisition frequency, and can obtain the environment monitoring data changing curve with less error compared with the high-frequency acquisition of fixed frequency.

## 1. Introduction

A portable meteorological station (PMS), in brief, is a kind of equipment that can automatically observe ground environment states, such as temperature, humidity, illumination and so on [1]. It is mainly used in many fields such as weather forecast [2], environmental monitoring [3] and biometeorology [4]. Generally, most PMSes use solar energy to store and supply power to the system when they are open, remote and the power grid is unable to supply power directly [5]. Continuous rainy or cloudy weather will lead to inefficient power storage of solar panels, thus affecting the efficiency of a PMS. 

Reducing the overall power consumption is very important for prolonging the life of a sensor and ensuring the stable operation of a PMS system [6]. Large capacity batteries are useful for prolonging the service time [7], but result in an increase in the cost and size of the PMS. In fact, reducing power consumption can also reduce electromagnetic radiation and thermal noise, and improve the stability of the system [8]. The existing low-power strategies usually depend on the design of low-power circuits [9], low-power sensors [10] and low-power chips [11], which can improve the performance of PMSes in hardware configuration [12]. However, it is not good enough in the practice of PMSes, and more efficient methods of energy management should be improved. Some studies have proposed a variety of acquisition frequency regulation methods to reduce power consumption. A harvesting and energy-aware adaptive sampling algorithm was proposed in [13]; it would set a low sampling rate and guarantee the self-sustainability when the residual energy of sensors is too low. Adaptive sampling technology for wireless sensors has been concerned recently. An adaptive sampling interval adjustment (ASIA) method using a two-input single-output fuzzy logic controller was proposed in [14], adaptive power management was proposed in [15] and an adaptive sampling algorithm based on temporal and spatial correlation of sensory data was proposed in [16]. An optimal power consumption control method of policy discrete-time queues was proposed in [17]. These algorithms consider the real-time state of the sensor and the external environment, so as to change the acquisition frequency of the sensor dynamically. 

In the view of statistics, many features of weather data have strong relationships with past days generally, especially with recent days. The influence of historical data on acquisition frequency can be considered. Therefore, this paper aims at modeling an acquisition interval mainly according to the recent changing rate of monitoring data. The main contribution of this paper is adjusting acquisition time intervals dynamically for wireless sensors by learning the changing tendency of monitoring data. By learning historical weather data with the genetic algorithm model, the acquisition frequency is dynamically adjusted to reduce power consumption.

## 2. Modeling on Historical Weather Data

### 2.1. The Acquisition Data Serials

The original data cited in this paper was collected from Breeding Base of Agasicles hygrophila in Yichang City, China. The base is located at Baishiping Village, Longzhouping Town, Changyang County. The geography coordinate of this location is E 111°13′22.7215″, N 30°31′33.7378″, and the altitude is 123.3 m.

The monitoring data is collected in the winter of 2018. It includes the air’s temperature, the air’s humidity and the sunlight’s strength. These environment monitoring data were collected by weather sensors.

The data is usually collected at some frequency [18], which is defined in monitoring protocol. Let us suppose a PMS collects *N* times of data at equal intervals every day. The weather acquisition serials of a certain day in *N* times are as follows *T* = {*T_1_, T_2_, …, T_N_*}. So, the interval of acquisition of wireless sensor is (60 × 24/*N*) minutes. The collected data is expressed as *y_t_*. The serials of data can be expressed as set *Y* = {*y*_1_, *y*_2_, …, *y_N_*}, especially the data serials of the *j*th day can be expressed as set *Y_j_* = {*y*_1*j*_, *y*_2*j*_, …, *y_Nj_*}. Data collected in this way are discrete, which can be fitted to a continuous environment state curve with many nonlinear fitting methods.

Since power consumption mainly depends on the quantity of data sent by sensors, reducing the sensor acquisition frequency is the main method of reducing energy consumption. The acquisition frequency can be adjusted by monitoring the center according to the requirements of information granularity, which takes effect on the control accuracy of the application system. Information granularity is decided by the number of sensor acquisition and intervals between adjacent acquisition periods. As for an application system, many data sent by sensors are redundant, since environment states monitored by sensors may change relatively little, causing little influence on information management. If as much redundant data as possible would not be sent by sensors, the energy consumption will decrease directly. 

As shown in Figure 1, a serial of feature points can be selected to form a fitted curve, which is quite similar to the initial curve formed by full data (the set *Y*) in high acquisition frequency. 

Obviously, the selected *Y′* is a sub-set of the set *Y*, that is *Y′*
⊆
*Y*. Coordinately, the selected acquisition time *T′* is a sub-set of set *T*, that is *T′*
⊆
*T*. Wireless sensors collect environment data according to the time-set *T′*, which is subject to a suitable distance *D* between the fitted curve formed by the collected data and the real curve. Figure 1 also shows that different fitted curves may be of quite different distance *D* even their numbers of selected data are the same. It means that a suitable time-set is most important.

### 2.2. Residual Sum of Square

The distance *D* mentioned above should meet the commands that an application system set for the control accuracy. For example, a greenhouse monitoring system may allow temperature fluctuation in several degrees. The distance *D* can be defined by the residual sum of square (RSS, in brief) of the *j*th day as: (1)$$D_{RSS_{j}}=\sum_{i=1}^{m}\frac{{\left(y_{ij}-{\hat{y}}_{ij}\right)}^{2}}{m}$$

Here, $y_{ij}$ presents the actual value from curve fitted from data in very high acquisition frequency of the *j*th day, whereas the ${\hat{y}}_{ij}$ is the predicted value of the *j*th day, it forms the optimized sequence. The lower the *D_RSS_* is, the more similar the two curves are.

Take Figure 1 as an example. The primitive curve is fitted by data collected every 10 min, 144 times one day in all. Its weather acquisition set is *T* = {*T*_1_, *T*_2_, …, *T*_144_}. Curve1 and curve2 in Figure 1 are fitted curves from data of different time serials *T′* and *T″*, respectively. 

The weather acquisition serials *T′* of cruve1 are as follows:$$T^{\prime }=\left\{\begin{array}{c} T_{2},T_{4},T_{10},T_{12},T_{18},T_{41},T_{53},T_{74}\\ T_{91},T_{114},T_{120},T_{128},T_{140},T_{143} \end{array}\right\}$$
and the weather acquisition serials *T″* of cruve2 are as follows:$$T^{\prime \prime }=\left\{\begin{array}{c} T_{2},T_{4},T_{5},T_{12},T_{23},T_{63},T_{64},T_{74}\\ T_{83},T_{91},T_{114},T_{124},T_{140},T_{143} \end{array}\right\}$$

Data sets are selected according to the same time-set from different fitted curves, and then the quality of different fitted curves can be compared. In the case of data in Figure 1, *D_RSS_* = 10.6989/144, curve1*D_RSS_* = 202.2039/144, curve2.

Obviously, the quality of the fitted curve is related to not only the quantity of selected acquisition data is but also the quality of selected data. If the number of selected acquisition data is too big, it will cause more energy consumption of wireless sensors. Conversely, it will not reflect the main features of the original environment state. The suitable sub-set of acquisition data should be found out to improve the quality of the fitted curve, which can be decided by an optimization model of suitable set-selection.

### 2.3. Optimization Model of Suitable Set Selection

#### 2.3.1. Analysis of Historical Data 

Take the temperature sequence as an example. Air temperature is affected by the strength of illumination, so the change of temperature is regular generally [19]. So, daily temperature changes of similar days have a similar trend. On the other hand, the time of sunrise and sunset also changes with the change of seasons, which leads to movement of some feature parts and change of maximum temperature on the temperature curves. Therefore, the fitted curves of similar days are always similar, and the fitted curves with a large time interval will be different.

The value of the Pearson Correlation Coefficient is usually applied to measure vector similarity, which is defined as:(2)$$r_{\left(j,j-1\right)}=\frac{\sum_{i=1}^{m}y_{ij}y_{i\left(j-1\right)}-\frac{\sum_{i=1}^{m}y_{ij}\sum_{i=1}^{m}y_{i\left(j-1\right)}}{m}}{\sqrt{\left(\sum_{i=1}^{m}y_{ij}^{2}-\frac{{\left(\sum_{i=1}^{m}y_{ij}\right)}^{2}}{m}\right)\left(\sum_{i=1}^{m}y_{i\left(j-1\right)}^{2}-\frac{{\left(\sum_{i=1}^{m}y_{i\left(j-1\right)}\right)}^{2}}{m}\right)}}$$
where *y_ij_* presents the data of the *j*th day in fitted curve, and the *y_i(j−1)_* presents the data of the (*j*−1)th day in fitted curve. The output range is [−1, 1]; 0 means no correlation, negative value means negative correlation, positive value means positive correlation.

Figure 2 shows the Pearson correlation coefficient for every two adjacent days of temperature data in a continuous week. It can be seen that the temperature data of adjacent days have an extremely strong correlation.

#### 2.3.2. Optimization Model Based on Adjacent Periods

The adjacent day’s original temperature data has strong relevance in Figure 2. The environment monitoring data changing curve is chosen from the previous days to the last day. The acquisition sequence of current acquisition data can be obtained with an optimization model according to the selected curve. The temperature variation tendencies of a similar duration in adjacent days are mostly similar, so it indicates the rules of acquisition time sequence of the next day.

So, the historical data of the past several days can be used to guide the acquisition sequence of the current day, assuming that the acquisition time-sequence of the current day is *T*. The purpose is to obtain a better sequence; in other words, the fitted curve from data in sequence *T* is more similar to the fitted curve from data in very high acquisition frequency. Generally, if the Pearson correlation coefficients of the past adjacent days are bigger than 0.8, multiple fitted curves of the past *M* days formed from the data in sequence *T* are similar.

The quality of fitted curves is defined by fitness function as follows:(3)$$FIT=\frac{\sum_{j=1}^{M}D_{RSS_{j}}}{M}=\frac{1}{M}\sum_{j=1}^{M}\sum_{i=1}^{m}\frac{{\left(y_{ij}-{\hat{y}}_{ij}\right)}^{2}}{m}$$
where $y_{ij}$ presents data set selected from the fitted curve of the *j*th previous day in a very high acquisition frequency and ${\hat{y}}_{ij}$ presents data set selected from the fitted curve of the *j*th previous day in sequence *T′*, where the total number of selected time points is *m*.

If the fitness value is large, it infers that the error between the fitted curve formed from data in sequence *T′* and curve from data in very high acquisition frequency may be large in the current day. Otherwise, *T′* is a better acquisition sequence. Whereas, when the fitness value *FIT* is getting smaller, the results come better. *T′* is limited by:(4)$$T^{\prime }\subseteq T$$

There will be some errors between the fitted curves formed by T′ and the real curve formed by T. The fitness value can reflect the error. In a different application field, the tolerance of error is different. Assume that:(5)$$FIT\leq FIT_{max}$$

The total number of optimized acquisition point in a selected day should not be too small or too big, since it is hard to reflect the main features of weather if the number is too small, or it wastes too much energy and the optimization steps are meaningless if the number is too big. So, the number *n* is limited by:(6)$$n\leq N,n\in N_{+}$$

In summary, the optimization model of adaptive algorithm for environment monitoring data acquisition strategy can be described as Equation (7):(7)$$\min FIT\;\mathrm{Subject}\;\mathrm{to}\;T^{\prime }\subseteq T\;FIT\leq FIT_{max}\;\mathrm{and}\;n\leq N,n\in N_{+}$$

## 3. Adaptive Genetic Algorithm of Frequency Adjustment

An adaptive genetic algorithm of optimizing acquisition frequency is proposed according to the optimization model mentioned above.

As for the genetic algorithm, both crossing operator *P_c_* and mutation operator *P_m_* have a great influence on the genetic algorithm’s performance, especially to the algorithm’s convergence. The population can produce new individuals when crossing operator *P_c_* is getting bigger, but if it becomes too large, the retention rate of the excellent individual in this population decreases. This algorithm is similar to the ordinary stochastic algorithm if mutation operator *P_m_* is too large, which means that the genetic algorithm is not needed. So, the adaptive genetic algorithm should be optimized.

### 3.1. Genetic Coding of Time Sequence Selection

The genetic algorithm cannot solve the parameter of the problem space directly. Therefore, it must be transformed into the genetic space problem, which lets genes make up the chromosome or the individual according to a certain structure. This transfer process is called coding or representation.

The genetic algorithm takes each acquisition serial as an individual. The genetic coding of every individual is a sequence, which is made up of N binary digits, where binary digit 1 represents data that is collected at this time point, while binary digit 0 represents data that is not collected at this time point. Therefore, the coding of weather acquisition serials T of a certain day contains N uniform acquisition frequency, and the coding of optimized acquisition serials T’ of a certain day contains n selected acquisition items.

Take Figure 1 as an example. The primitive curve is fitted by data collected every 10 min, i.e., 144 times one day. Its coding is:{*T_i_* = 1|*i* = 1, 2, …*N*}

Curve1 and curve2 in Figure 1 are fitted according to their own acquisition strategy. The coding of acquisition strategy:{*T′_i_* = 0 or 1|*i* = 1, 2, …*N*}

The initialization of population is completed by adopting random initialization. The initialization steps of every individual are:Generates a zero vector of length *N*.Assigns the first *n* elements to binary 1.Disorganizes those elements order randomly.Obtains the coding of individual, which has *n* binary 1 and *N−n* binary 0.

Each initialization step produces a genetic individual randomly, and all generated individuals form the initial population.

### 3.2. Fitness Function

Fitness of genetic algorithm describes the environmental adaptability of every individual, the larger the adaptability, the higher the chance of survival. Therefore, there are some acquisition sequences for the individuals that do not meet the requirements of acquisition frequency, the total acquisition number $sum\left(T_{i}\right)\neq n$. If the individual’s fitness value is 0, it means a wrong acquisition. For a feasible individual, the reciprocal of the order function can be used as the fitness function. So, the fitness function can express as:(8)$$F=\{\begin{array}{c} \frac{1}{FIT},sum\left(T_{i}\right)\neq n\\ 0,\mathrm{else} \end{array}$$

### 3.3. Selection Operation

Selection operations of the genetic algorithm include choosing the better individuals from the population and giving up the bad ones. The purpose of selection is to transfer the better individuals to the next generation. In this way, the algorithm optimizes the average fitness of each generation. The selection operation uses the wheel selection algorithm normally.

### 3.4. Crossover Operation

The crossover operator of the genetic algorithm is an important role, which helps to improve the searching ability of the genetic algorithm. A larger crossover probability is chosen to enrich the species diversity and improve the searchability in the beginning, while reducing the crossing probability to protect the best species from damage at the end. The genetic operators *P_c_* can be defined as:(9)$$P_{c}=\{\begin{array}{cc} P_{c1}-\frac{\left(P_{c1}-P_{c2}\right)\left(f^{\prime }-f_{avg}\right)}{f_{\max}-f_{avg}} & ,f^{\prime }\geq f_{avg}\\ P_{c1} & ,f^{\prime }<f_{avg} \end{array}$$
where, *P_c_*_1_ and *P_c_*_2_ are constant, *f_max_* is the maximum fitness value of the population, *f_avg_* is the average fitness value of each generation and *f*′ is the larger fitness value of the two individuals to the crossed.

In order to ensure the number of the individual’s gene 0 and gene 1 be constant after the crossover operations, the crossover operations can be executed at the same time if the numbers of diffident genes on the left and right sides of two individual intersections are the same. If it is not the same, it must be changed to the same with moving the position of the intersection and then execute the crossover operation.

### 3.5. Mutation Operation

The mutation operation of the genetic algorithm is to replace a genotype gene of the individual chromosome-coding string with other alleles. In this way, a new individual can be obtained. Thus, the genetic algorithm has the ability of local search, maintains species diversity and prevents premature convergence. Similar to the crossover operation, the mutation probability is defined as:(10)$$P_{m}=\{\begin{array}{cc} P_{m1}-\frac{\left(P_{m1}-P_{m2}\right)\left(f_{\max}-f\right)}{f_{\max}-f_{avg}} & ,f\geq f_{avg}\\ P_{m1} & ,f<f_{avg} \end{array}$$
where, *P_m_*_1_ and *P_m_*_2_ is constant, *f_max_* is the maximum fitness value of the population, *f_avg_* is the average fitness value of each generation and *f* is the fitness of the mutated individual.

In order to ensure the proportion stability of gene 0 or gene 1, two mutation points are generated. If genes of the two mutation points are different, the gene of the mutation points will occur simultaneously, and vice versa.

### 3.6. Error Correction

There exists an error between the fitted curve and the real curve. It will result in an accumulative error if the new day’s acquisition sequence is calculated only based on the fitted curve formed the previous day. So, the error should be corrected by applying high acquisition frequency every several days.

The setting of correction days should be based on the type of data collected and the maximum allowable error. If it is too small, the average acquisition frequency will increase and the power consumption will increase too. If it is too large, the error of data acquisition will be too large accordingly.

### 3.7. Main Algorithm Process

The flow of the main algorithm is shown in Figure 3.

The main algorithm processes mainly include:Step 1: Parameter initialization, such as the allowed error *FIT_max_*, acquisition frequency *n_init_*, iterations number of adaptive genetic algorithm i, interval-days *C* of correcting error.Step 2: Judging whether the current day is error correcting day or not; if it is the day of error correcting, go to Step 6, else go to Step 3.Step 3: Generating the initial population which contains *p* individuals randomly.Step 4: Obtaining a new population with a serial of operations such as: selection operation, crossover operation and mutation operation.Step 5: If the number of iterations does not reach the maximum number of iterations, repeat Step 4. Otherwise, if the individual’s maximum fitness of the new population is not satisfied with the maximum number of iterations, the acquisition frequency *n* should be adjusted higher, then go to Step 3. The current data acquisition strategy returns. Then go to the Step 7.Step 6: Taking the acquisition sequence *T* in high frequency as the current acquisition strategy, and the initial acquisition number *n_init_* as acquisition number n.Step 7: Obtaining data according to the above acquisition sequence and updating the fitted data of current day.

## 4. Experiment Analysis

### 4.1. Effect Analysis

In order to testify the performance of the proposed algorithm, experimental sensor data in high acquisition frequency is collected for one month, which is taken as the test dataset. The optimized acquisition sequence is produced by the proposed algorithm in Section 3, and the quality of the result is compared to the test dataset.

Taking the temperature data in our project as an example, the data is collected *N* = 144 times every day if the data is not optimized. According to the previous three days’ environment monitoring data changing curve, the current day’s acquisition sequence can be obtained. In this experiment, *M* = 3, initial acquisition frequency *n_init_ =* 20, tolerable error *FIT_max_* = 0.5, Iteration times *i* = 100, population size *p* = 1000, correction days *C* = 7. *P*_*c*1_ = 0.9, *P*_*c*2_ = 0.6, *P*_*m*1_ = 0.1, *P*_*m*2_ = 0.01. Different fitting algorithms have a certain influence on the error. Because there are few data points, the fitting algorithm uses the least square method to eight degree fit function (which depends on the scale of acquisition frequency; it should not be too big if the scale is small. Here, eight degrees is selected for the experiment data on the balance of the fitting accuracy and computation cost). The comparison of some part of fitted curves is shown in Figure 4. The algorithm applies to get the acquisition strategy. The low-frequency data from the optimized acquisition strategy is compared with the actual data.

The result quality is usually evaluated with four evaluating indicators, RMSE, MAE, MAPE and Pearson correlation coefficient (r), which are defined as follows:(11)$$\mathrm{RMSE}=\sqrt{\frac{1}{m}\sum_{i=1}^{n}{\left(y_{i}-{\hat{y}}_{i}\right)}^{2}}$$
(12)$$\mathrm{MAE}=\frac{1}{m}\sum_{i=1}^{m}\left(\left|y_{i}-{\hat{y}}_{i}\right|\right)$$
(13)$$\mathrm{MAPE}=\frac{1}{m}\sum_{i=1}^{m}\left(\frac{\left|y_{i}-{\hat{y}}_{i}\right|*100}{y_{i}}\right)$$
(14)$$r=\frac{\sum y_{i}{\hat{y}}_{i}-\frac{\sum y_{i}\sum {\hat{y}}_{i}}{m}}{\sqrt{\left(\sum y_{i}^{2}-\frac{{\left(\sum y_{i}\right)}^{2}}{m}\right)\left(\sum {\hat{y}}_{i}^{2}-\frac{{\left(\sum {\hat{y}}_{i}\right)}^{2}}{m}\right)}}$$

$y_{i}$ is the value in real line, ${\hat{y}}_{i}$ is the value in modeling line and *m* is observed sample number.

The first three items indicate quality of optimized results. Take their normalized mean as an integrated evaluation parameter:(15)$$e=\frac{\left(RMSE+MAE+MAPE\right)}{3}$$

After testing data of one month, Figure 5 shows that the errors of optimized results are lower than 1 °C. On the other hand, the Pearson correlation coefficient presents linear relationships with the real observe value. The results in Figure 6 show that most of correlation coefficients are bigger than 0.8, which means very strong correlation.

### 4.2. Influenced Factors Analysis

Let the initial acquisition frequency *n*_init_ = 14 times, other conditions are same as before, and let *FITmax* = 1.0, *FITmax* = 0.75 and *FITmax* = 0.5. Figure 7, Figure 8 and Figure 9 shows their comparisons, respectively.

The comparison results are listed in Table 1. If the data accuracy is strictly required, a higher *F_min_* value needs to be set. Meanwhile, when the *F_min_* rises up, the algorithm itself needs a higher acquisition frequency *n* in order to reach higher acquisition accuracy. If the initial acquisition time is large, it will cause too large acquisition frequency, and it will cost too much power. On the other hand, if the initial acquisition is low, a large number of iterations will be needed; this will cost too much time. When the is *F_min_* fitted, choose the correct *n_init_* that is good for the algorithm progress.

The local original data will help us to find the correct data *n*_init_. Many experiments have been carried out to evaluate the acquisition frequency *n*, and the mean error, maximum error and minimum error are compared.

The result in Figure 10 shows that there is a negative correlation between *n* and *F_avg_*. On the direction of this curve, the correct *n_init_* can be chosen to improve the algorithm performance.

### 4.3. Convergence Performance Comparison of Different Algorithms

Convergence performance is very important for a genetic algorithm. Figure 11 gives the comparison between normal genetic algorithm (GA, in brief) and adaptive genetic algorithm (AGA, in brief) proposed in the paper. It shows that AGA is quite beneficial than the normal GA when the iterations are less than 80, and the fitness converges to the same level.

### 4.4. Adaptive Analysis of Other Weather Data

The algorithm can also be applied to evaluate the humidity and illumination values. Similar to the processes of temperature data, the low-frequency acquisition data and the real data can be obtained, and comparison results are shown in Figure 12 and Figure 13.

It shows that the algorithm has good adaptive capacity for handling monitored environment data with a high Pearson correlation relationship.

## 5. Conclusions

This paper proposes an adaptive adjustment method of environment monitoring data acquisition strategy for reducing the power consumption of PMSes. This method is based on the adaptive genetic algorithm. By optimizing the daily acquisition strategy, a suitable weather curve can be fitted by data in very low acquisition frequency. The experimental results under the actual environment show that the algorithm can effectively optimize the acquisition strategy.

## Figures and Tables

**Figure 1 sensors-20-00990-f001:**
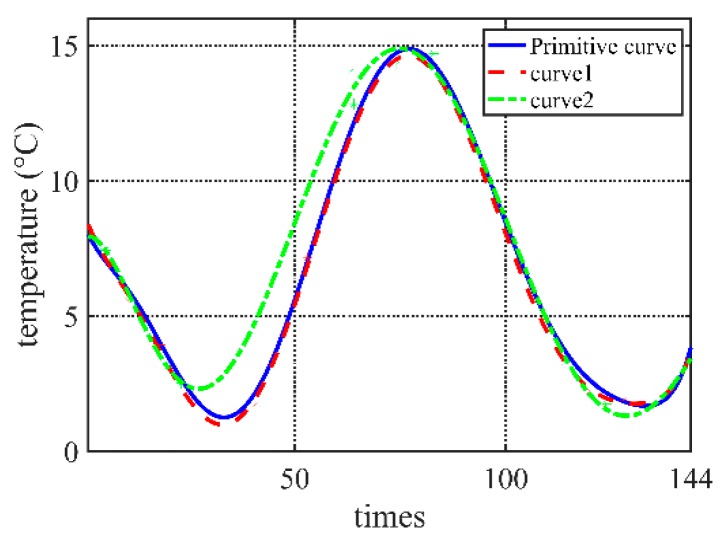
Comparison of fitted curves in different acquisition frequencies.

**Figure 2 sensors-20-00990-f002:**
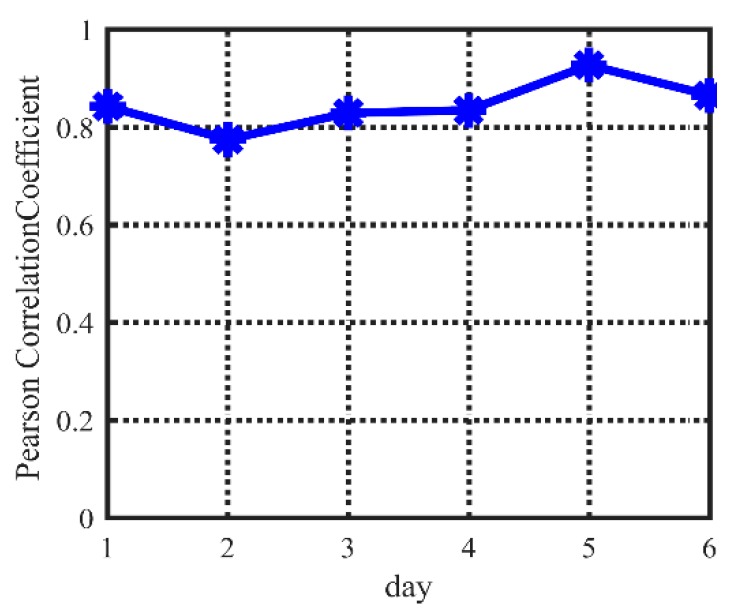
The adjacent day’s Pearson coefficient.

**Figure 3 sensors-20-00990-f003:**
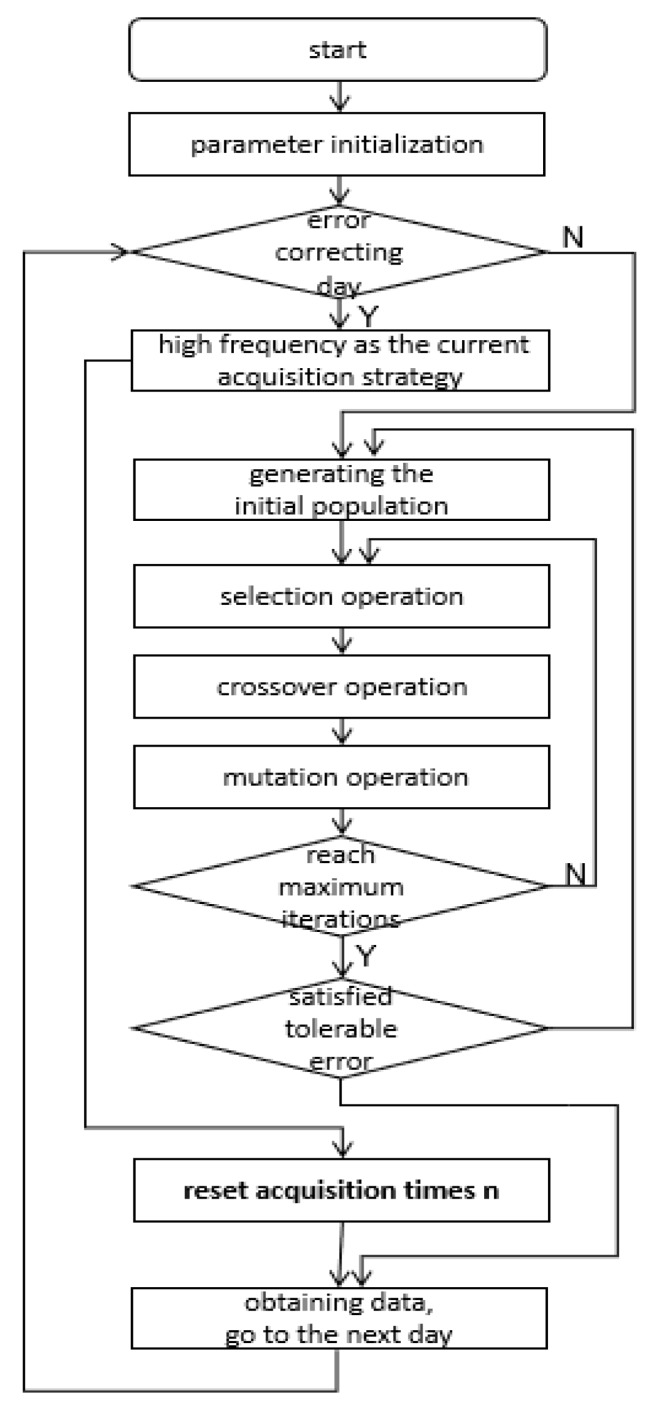
Main flow.

**Figure 4 sensors-20-00990-f004:**
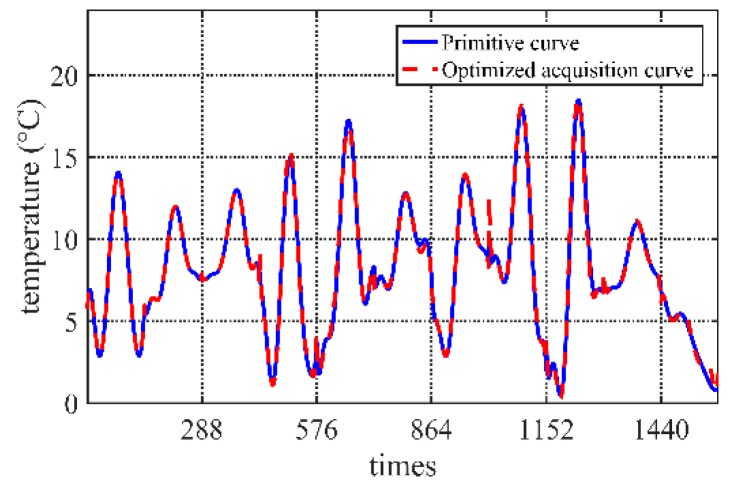
Comparison of optimized curve and primitive curve.

**Figure 5 sensors-20-00990-f005:**
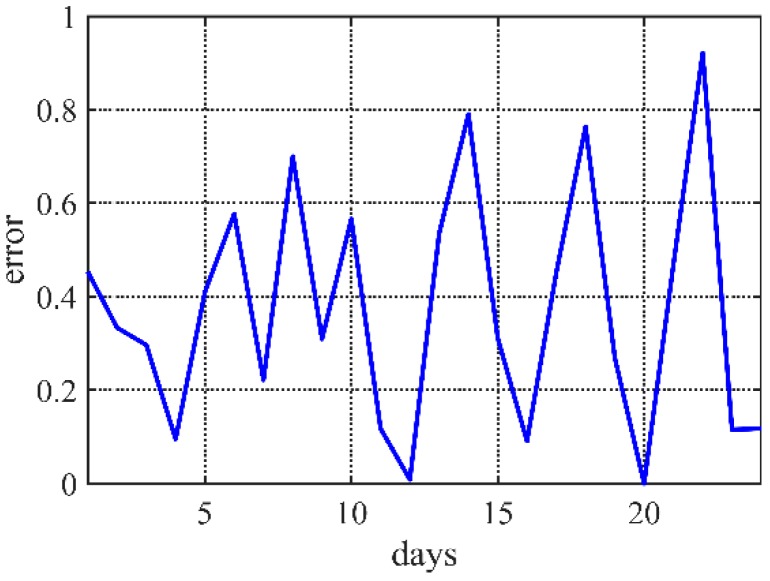
The error of several days.

**Figure 6 sensors-20-00990-f006:**
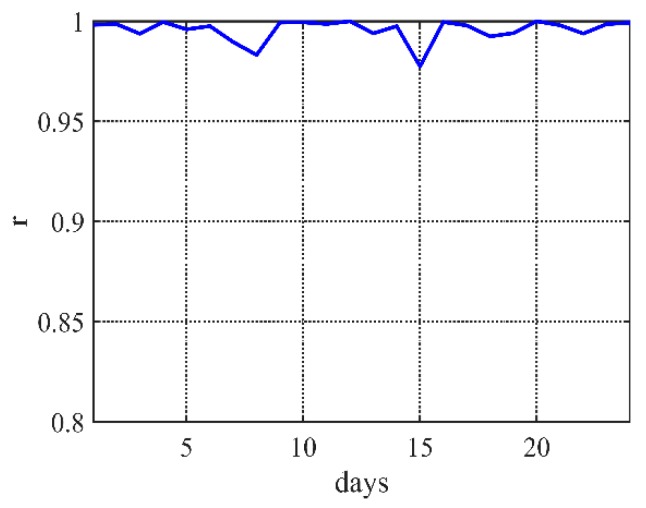
The Pearson correlation coefficient of several days.

**Figure 7 sensors-20-00990-f007:**
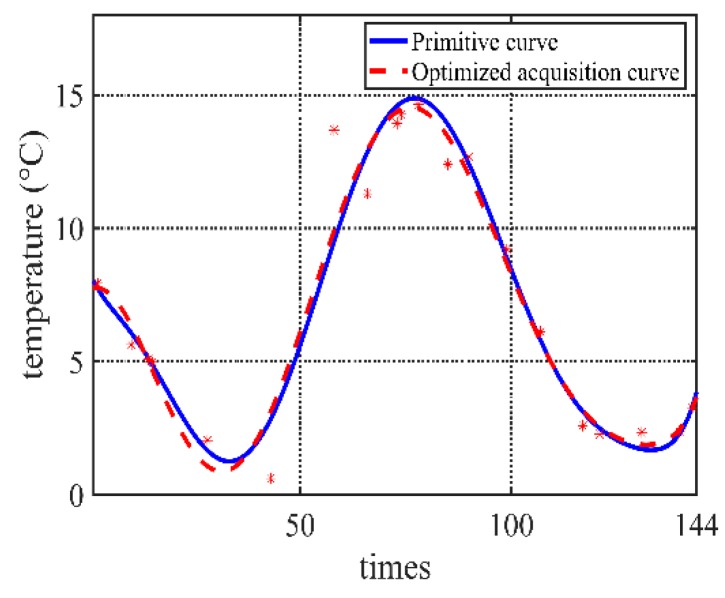
Comparison when *FIT_max_* = 0.5.

**Figure 8 sensors-20-00990-f008:**
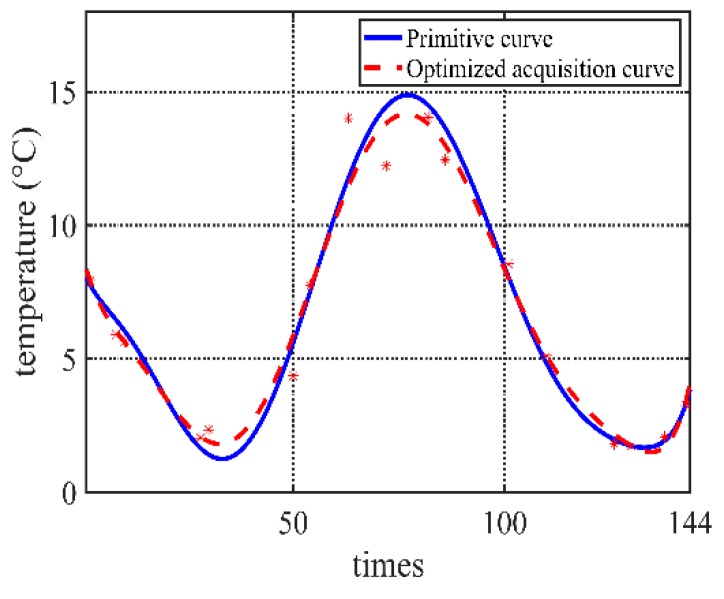
Comparison when *FIT_max_* = 0.75.

**Figure 9 sensors-20-00990-f009:**
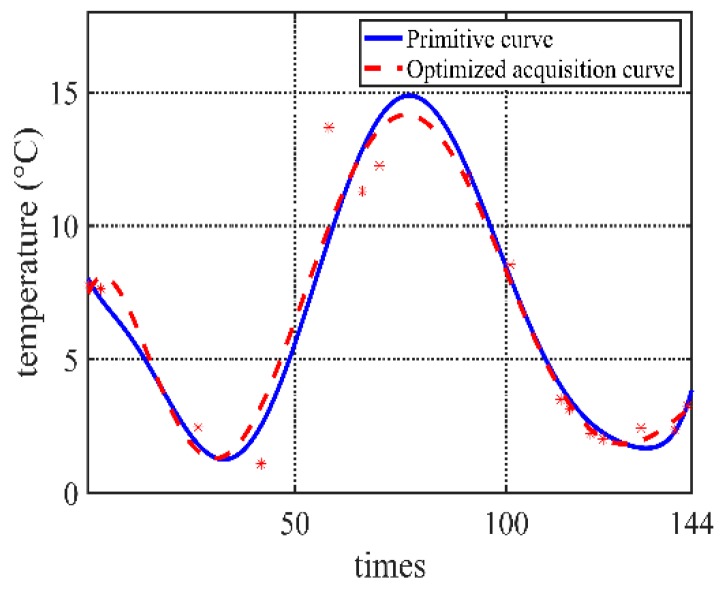
Comparison when *FIT_max_* = 1.0.

**Figure 10 sensors-20-00990-f010:**
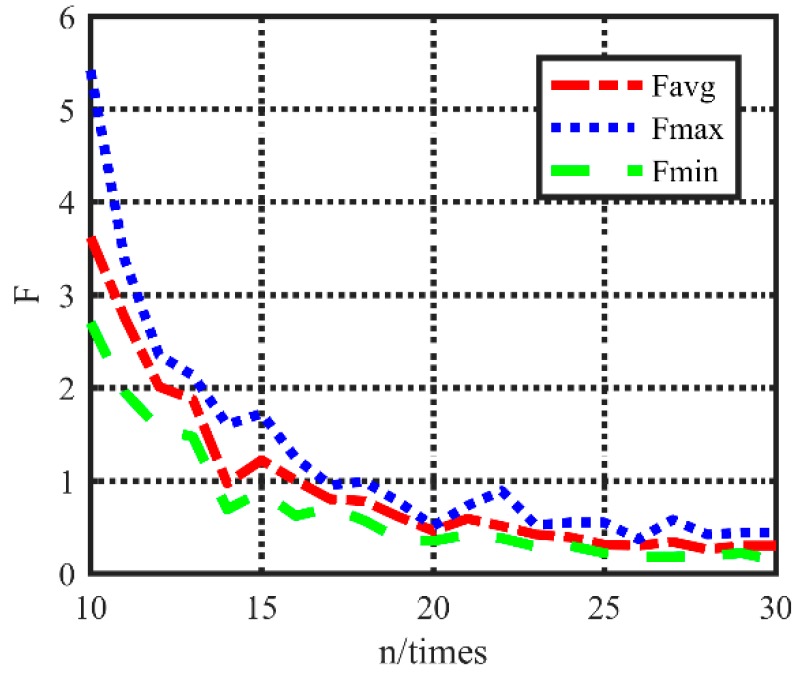
Acquisition frequency vs. error value’s mean value.

**Figure 11 sensors-20-00990-f011:**
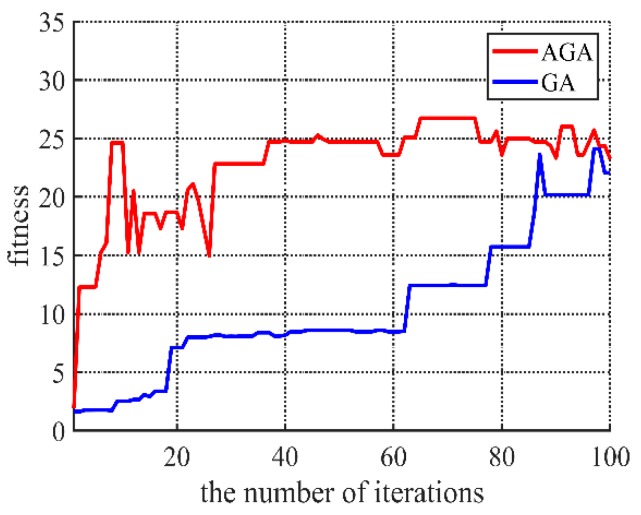
Convergence rate of adaptive genetic algorithm (AGA) and genetic algorithm (GA).

**Figure 12 sensors-20-00990-f012:**
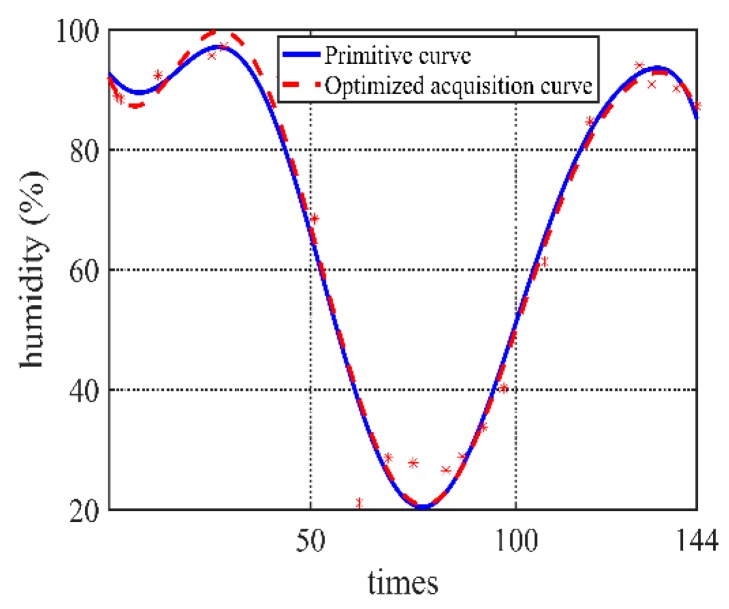
Algorithm humidity collection and the real data.

**Figure 13 sensors-20-00990-f013:**
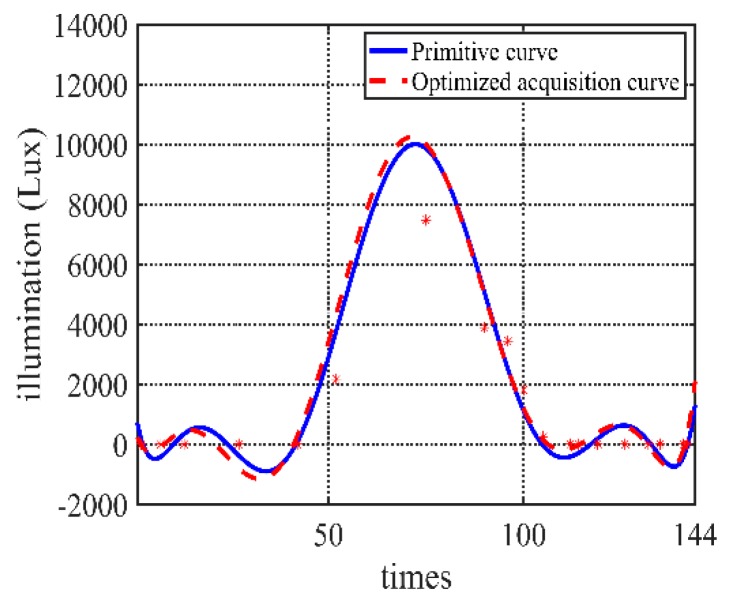
Algorithm illumination value acquisition and the real data.

**Table 1 sensors-20-00990-t001:** Different *FIT_max_*, acquisition frequency and error.

*FIT_max_*	Acquisition Frequency	RMSE	MAE	MAPE	r
1.00	16	0.509	0.418	9.463	0.995
0.75	17	0.397	0.343	8.954	0.998
0.50	20	0.284	0.242	6.125	0.998

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
