# Peer review of "An Adaptive Genetic Algorithm of Adjusting Sensor Acquisition Frequency"

_sensors, 2020, doi:10.3390/s20040990_

Round 1

Reviewer 1 Report

The paper is more like a working report for the data collected. It lacks sufficient scientific content and engineering innovation. 

There are some misspellings and unclear sentences.

It is not justified why degree 8 polynomial is enough for the environment monitoring data curve fitting and why historical data is of use for the prediction

in the climate change era.

Author Response

Dear Reviewer,
    Thank you for your kindly comments. We have revised it according to suggestions you and other reviewer mentioned. The main modifications are:
    1) The Grammar is checked thoroughly, and the final polish of whole context is given. 4 valuable references added, and more related literature survey is modified.
    2) All formatting are checked according to the sensors template, including the reference list style
    3) As for the reason that degree 8 polynomial is selected for climate curve fitting, the degree is selected depending on the scale of acquisition frequency, it should not be too big if the scale is small. According to our experience, the practical frequency is less than 20 generally, it will waste computation cost with less accuracy improvement if the degree is more than 8. It is not the main work of this paper, so only a short illustration is given in continued context.

Reviewer 2 Report

The authors proposed a model of adaptive genetic algorithm for environment monitoring data acquisition strategy.

Weak Points:

Novelty of the study are not clear. Advantages of the proposed model is not clear. Literature survey is not adequate. Figures, inset explanations and captions are not clearly visible. Grammar and formatting should be developed. Reference list style are not suitable to the Sensor manuscript template.

Author Response

Dear Reviewer,
    Thank you for your kindly comments. We have revised it according to suggestions you and other reviewer mentioned. The main modifications are:
    1) 4 valuable references added, and more related literature survey is modified.
    2) The Grammar is checked thoroughly, and the final polish of whole context is given.
    3) All formatting are checked according to the sensors template, including the reference list style
    4) This paper aims at modeling on acquisition interval mainly according to the recent changing rate of monitoring data. The main contribution of this paper is adjusting acquisition-time intervals dynamically for wireless sensors by learning the changing tendency of monitoring data. The optimization model is based on the knowledge that the most tendency of monitored items in suitable periods changes linearly. It is valuable for outdoors sensor applications.

Round 2

Reviewer 1 Report

Maybe fitting results employing polynomials other than 8 can be reported to show 8 is an appropriate choice.

Author Response

Dear Reviewer,

    Thank you for your kindly comments. We have made further revision according to suggestions you and other reviewer mentioned. As you suggested, degree-8 polynomial may not be the best choice. It depends on the scale of acquisition frequency, it should not be too big if the scale is small. Eight-degree is selected for the experiment data on the balance of the fitting accuracy and computation cost in this paper, for which we have made some comparison experiment. Since that all fitting curve applies the eight-degree polynomial, it has less influence on evaluating our model. So, we don’t give more illustration in continued context.

Reviewer 2 Report

Thank you for explaining the "aim of the manuscript" to the Reviewer.  However, I recommend you to add this information to the end of the Introduction section of the manuscript.

Author Response

Dear Reviewer,

    Thank you for your kindly comments. We have made further revision according to suggestions you and other reviewer mentioned. As you suggested, a little revision is given at the end of introduction part.
